# Population size, density, and ranging behaviour in a key leopard population in the Western Cape, South Africa

**Lana Müller**[1]*, **Willem Daniel Briers-Louw**[2], **Barbara Catharine Seele**[1], **Christiaan Stefanus Lochner**[1], **Rajan Amin**[3]

**1** The Cape Leopard Trust, Tokai, Cape Town, South Africa, **2** Zambeze Delta Conservation, Marromeu, Mozambique, **3** Conservation Programmes, Zoological Society of London, Regent's Park, London, United Kingdom

* lmuller1985@gmail.com

**Data Availability Statement:** The data relevant to this study are available in the OSF repository at osf. io/etn9r.

## Abstract

Globally, leopards are the most widespread large felid. However, mounting anthropogenic threats are rapidly reducing viable leopard populations and their range. Despite the clear pressures facing this species, there is a dearth of robust and reliable population and density estimates for leopards across their range, which is particularly important in landscapes that consist of protected and non-protected areas. We conducted a camera trapping survey between 2017 and 2018 in the Western Cape, South Africa to estimate the occupancy, density, and population size of a leopard population. Leopards were recorded at 95% of camera trapping sites, which resulted in a high occupancy that showed no significant variation between seasons, habitat types, or along an altitudinal gradient. Our results indicated a low leopard density in the study area, with an estimated 1.53 leopards/100 km² in summer and 1.62 leopards/100 km² in winter. Mean leopard population size was therefore estimated at 107 and 113 individuals in the winter and summer respectively. Leopard activity centres for female ranges were centred in the core study area and could be predicted with good certainty, while males appeared to move out of the study area during winter which resulted in a higher uncertainty in locations of activity centres. Interestingly, livestock depredation events in the surrounding farmlands were significantly higher in winter, which coincides with male leopards moving outside the core protected area into the surrounding farmlands. To reduce livestock losses and retaliatory leopard killings, we suggest that human-carnivore conflict mitigation measures be intensely monitored during the winter months in the study area. We also suggest that future leopard conservation efforts should focus on privately-owned land as these non-protected areas contain the majority of the remaining suitable leopard habitat and may provide important dispersal corridors and buffer zones on which the long-term sustainability of leopard populations depends.

**Funding:** The work is supported by the Cape Leopard Trust whom received funding from Abax Foundation, Bushmans Kloof Wilderness Reserve, Lomas Wildlife Protection Trust, and the Rolf Stephan Nussbaum Foundation. Ford Wildlife Foundation has sponsored a vehicle for the research team.

**Competing interests:** The authors have declared that no competing interests exist.

## Introduction

Leopards are the most widespread felid worldwide, ranging across much of Africa and tropical Asia [1]. However, in the past century their range has declined globally by 63–75% and in Africa by 48–67% [2] with the most significant range loss having occurred in North and West Africa [2, 3]. As a result, leopards were up-listed to Vulnerable on the IUCN Red List in 2016 and have maintained this status due to increasing anthropogenic threats [3]. Habitat loss and fragmentation, prey-base depletion due to illegal bushmeat poaching, and retaliatory killing as a result of livestock depredation are the main drivers of leopard population decline across their range, while unsustainable trophy hunting and poaching for body parts also pose major threats to leopards [3–7].

In South Africa, approximately 20% of the country is considered suitable leopard habitat, with only one third of this habitat falling within protected areas [8]. Their distribution is highly fragmented and has been grouped into four core regions: 1) west and southeast coast of Western and Eastern Cape Provinces, 2) interior of KwaZulu-Natal Province, 3) Kruger National Park and interior of Limpopo, Mpumalanga and North West Provinces, and 4) a northern region, containing the Kgalagadi Transfrontier Park and adjacent areas of the Northern Cape and North West Province [8]. Each of these subpopulations contain fewer than 1,000 mature individuals, with the exception of the Kruger National Park and the interior of Limpopo, Mpumalanga, and North West Provinces [8]. Current models suggest that the national leopard population is likely to suffer further declines, particularly outside protected areas where leopards are at risk due to snaring, poisoning, problem animal control, and unregulated trophy harvesting [3, 8].

In the Western Cape, leopards are the last remaining large carnivore after other large carnivores such as spotted hyena (*Crocuta crocuta*) and lion (*Panthera leo*) were extirpated several centuries ago [9, 10]. However, leopards in the Cape have suffered substantial long-term persecution and habitat loss [11]. Leopards were considered 'vermin' until 1968 by the Administration of the Cape Province and it was only in 1974 that they were declared a "protected wild animal" in the province, which meant that a permit was required to trap or shoot a leopard (Nature Conservation Ordinance No. 19 of 1974) [12, 13]. Despite these changes in legislation, persecution of leopards continues due to livestock depredation on privately-owned farms [11, 12, 14]. Their persistence in the province is largely due to the protection of rugged mountain landscapes, which provide important refuges, as well as their adaptability within the human-dominated landscape.

Throughout the Western Cape Province there is approximately 50,000 km$^2$ of potential leopard habitat remaining, with only 15,000 km$^2$ within conservation areas and mountain catchment zones, and the other 35,000 km$^2$ within privately-owned farmland and surrounding small towns [8, 15]. In the province, leopards appear to have one of the lowest densities in the country, ranging from 0.25–2.3 individuals/100 km$^2$, and are also known to occupy large home ranges (35–910 km$^2$) [11, 16] which expose leopards to detrimental edge effects, especially if they move beyond the borders of the protected areas [17]. Given their range and habitat restrictions as well as conflict with humans, it is essential to better understand leopard spatial distribution, density, and trends within the region to facilitate informed decision-making to improve conservation efforts for the species [18].

Currently, there is a paucity of robust density estimates for leopards across their range. However, inappropriate attempts to estimate leopard populations on a broad-scale [19, 20] may also result in more harm than good, as critiqued by Norton [21]. For example, the issuing of problem animal control permits and trophy hunting quotas are typically based on overestimated leopard densities [21, 22]. Such inaccuracies can lead to poor management

recommendations [23] and may have major implications for leopard conservation [17, 22], such as the incorrect assumption that the leopard conservation status is assured [24]. Thus, it is crucial that research efforts prioritize the collection of robust density estimates to improve management and conservation of this charismatic species [6, 22, 25, 26].

Efforts have previously been made to provide population estimates for leopards in the Cederberg region of the Western Cape. However, these studies were limited to the radio collaring of a few individuals [11, 27], spoor counts [28], as well as less robust camera trapping methods [11], which provided a wide range of imprecise estimates. The main aim of this study was to provide robust estimates of the density and distribution of a leopard source population in the Western Cape of South Africa. Furthermore, we provide seasonal leopard range patterns and assess its implication on livestock depredation.

## Methods

### Study area

The study was conducted in the Cederberg mountains (32˚27'S; 19˚25'E) in the Western Cape, South Africa (Fig 1). This area lies approximately 200 km north of Cape Town and slightly east of the towns of Clanwilliam and Citrusdal [29]. The area covers approximately 3,000 km$^2$ of rugged mountainous terrain of which 1,500 km$^2$ are protected. The protected area includes Matjiesrivier Nature Reserve, the Cederberg Wilderness Area, the Cederberg Conservancy,

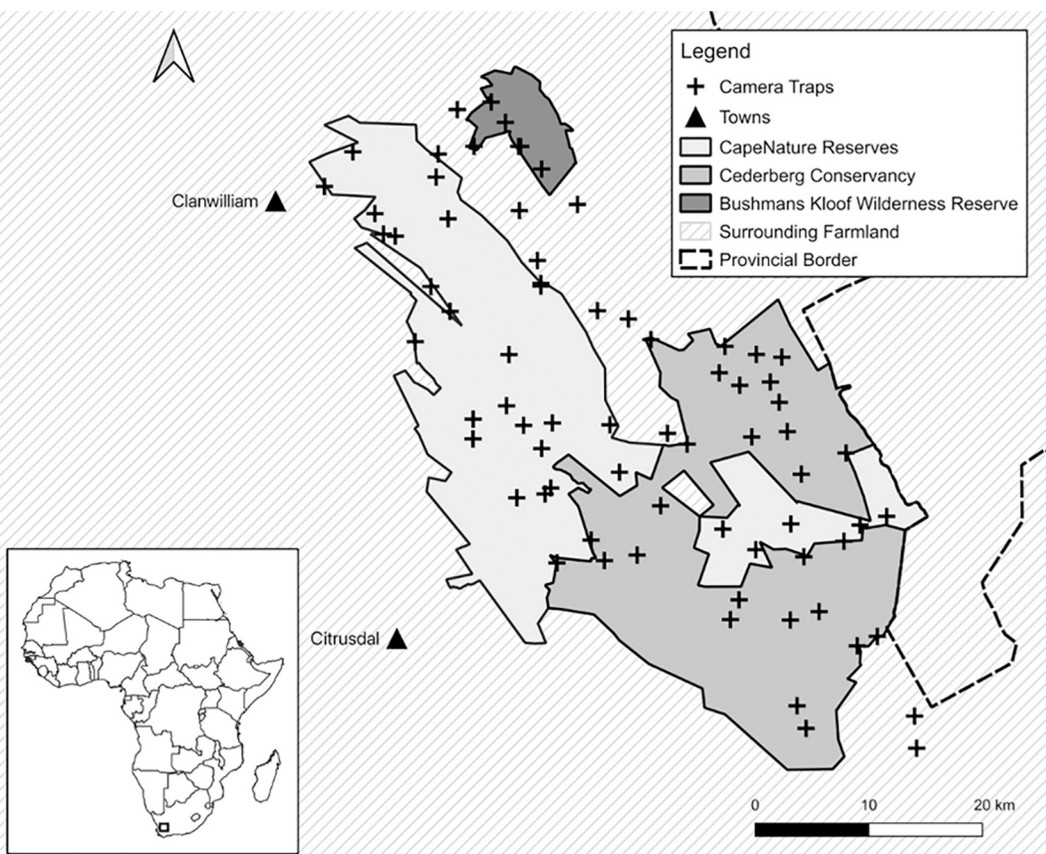

**Fig 1. Camera trapping locations (n = 73) within the Cederberg study area.** Inset: Map of Africa and the location of the study area indicated with a black square.

and Bushmans Kloof Wilderness Reserve. The remaining area primarily consists of privately owned farms. Up until the 1990's, livestock farming represented the main land use in the area [11] and the Cederberg was regarded as one of the largest farmer-leopard conflict hotspots in the Western Cape [27]. Since then, other farming practices such as wine, citrus, olive, and tea (rooibos and buchu) have become more common. Tourism has also increased in the area due to attractions such as camping, hiking, rock climbing, and mountain biking.

The semi-arid Cederberg landscape lies within the Cape Floristic Kingdom and is home to several endemic plant species such as the Clanwilliam cedar (*Widdringtonia cedarbergensis*), the snow protea (*Protea cryophile*), and the red rocket pincushion (*Leucospermum reflexum*) [30]. Furthermore, the Cederberg also represents an important intersection of two biomes, with the Fynbos Biome in the west consisting of fynbos (restioids, ericoids and proteoids) and renosterveld vegetation (Renosterbos spp.), and the Succulent Karoo biome in the east comprising of small shrubs and succulents [31]. The topography in both biomes consists of rugged sandstone and shale mountains divided by valleys and ravines that are either densely wooded or open [29]. In the fynbos section, altitude ranges from 200 to 2026 m, while in the Karoo section it varies from 258 to 1446 m [29].

The Cederberg has a Mediterranean climate with winter from April to September and an austral summer from October to March [29]. Mean temperatures range from 22 to 40˚C in summer and 10 to 15˚C in winter, with sporadic snowfall on the higher mountain peaks [32]. Conditions can be harsh with extreme temperatures reaching up to 47˚C in summer and below freezing (–7˚C) in winter [29]. Rainfall primarily occurs during winter with occasional thunderstorms in summer. The wetter Fynbos Biome has a mean annual rainfall of 669 mm, while the drier Karoo biome has a mean annual rainfall of 179 mm [29].

## Camera trapping and data processing

A permit (No: 0056-AAA007-00217) for camera trapping and scat collection was obtained from CapeNature, the governing body of nature conservation in the Western Cape Province of South Africa prior to the survey. We selected 73 camera trap stations distributed across the study area from October 2017 to September 2018, covering an effective sampled area of 2,823 km$^2$ (Fig 1, S1 Table). The standard protocol for selecting a suitable trapping grid is using the minimum home range size of the target species which reduces the likelihood of potential gaps where individuals could be missed in the effective survey area [33]. Based on a previous study in the Cederberg, mean home range sizes for females was 117±29 km$^2$ and males was 313±108 km$^2$, whilst the minimum recorded female home range size was 74 km$^2$ [11]. Therefore, we divided the area into conservative grid cells of 50 km$^2$. Each grid cell contained two camera trap stations to ensure sufficient coverage of all potential leopard home ranges. Mean inter-camera trap station distance was 2.78 km (range = 0.5–6 km) which satisfies the assumption that no animal had a zero probability of being captured [33].

Cameras were placed on hiking trails, game trails, jeep tracks, or along natural features (i.e. gorges or drainage lines) where leopards were likely to move, and where signs of leopards were present (e.g. scat, tracks, or scratch trees) to maximise the likelihood of capturing leopards. A pair of white-flash cameras (Cuddeback X-Change Color Model 1279) were placed 2–3 m from the trail, with the cameras placed on either side of the trail. Camera traps were positioned perpendicular to the trail at a height of approximately 40 cm above the ground to obtain full lateral body images of passing animals [11]. As the rosette patterns of a leopard are asymmetrical on either side of the body, paired cameras allowed for a simultaneous left and right flank image to facilitate the accurate identification of individuals. Cameras were set to operate 24 hours a day, to take three photo bursts every time the sensor was triggered with a minimum

trigger delay of 1/4 of a second. The flash strength setting varied between the "indoor" setting for narrow trails to the "close" setting for wider jeep tracks. Vegetation was pruned to the ground in front of each camera to limit false triggers. Cameras were serviced (batteries replaced, images downloaded, and vegetation pruned, if necessary) once every two months. Images were processed using *Camera Base* version 1.3 [34]. All animals in the photographs were manually identified to species level, while leopards were identified to individual level using the pattern recognition program *HotSpotter* [35]. Individual leopards were subsequently manually verified and compared to the Cederberg leopard database. A threshold of 30 minutes was used to temporally distinguish independence of unique leopard photo-capture events [36].

## Population size and density

Leopard population densities were estimated using Bayesian spatial capture-recapture (SCR) models. SCR models are relatively new methods which extend traditional capture-recapture models by incorporating auxiliary information about individual capture locations into the modelling framework to explicitly estimate the density of animal activity centres, with animal abundance as a derived parameter [37, 38]. To ensure population closure, the summer (1 October 2017–31 March 2018) and winter (1 April 2018–30 September 2018) month datasets were analysed separately (Fig 2).

Leopards exhibit sex-specific differences in space-use and behaviour, with the home range of a single adult territorial male overlapping with smaller home ranges of several females [39, 40]. Location of cameras is also a likely source of capture heterogeneity [41, 42]. The model p0

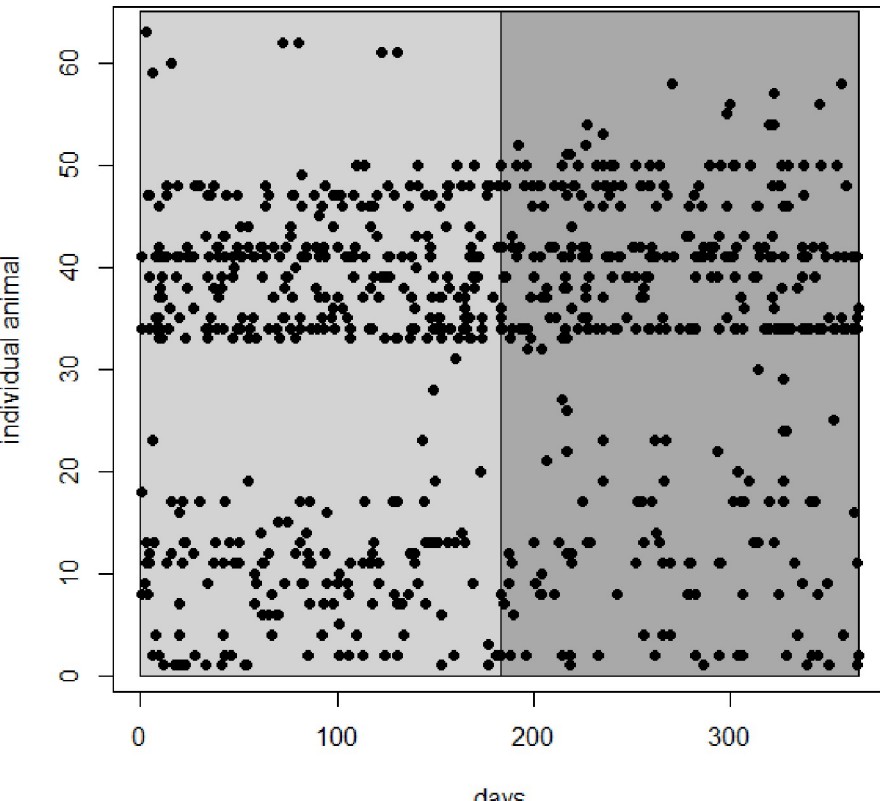

**Fig 2. Individual leopard detections over the summer (light grey) and winter (dark grey) of 2017–2018, in the Cederberg, Western Cape.**

(sex+trap).sigma(sex), where p0 denotes the probability of capture when the distance between the animal's activity centre and the camera is zero and sigma is the ranging scale parameter, was implemented in the program JAGS (Just Another Gibbs Sampler) accessed through the program R, version 4.0.4 [43] using the package RJAGS (http://mcmc-jags.sourceforge.net). In data augmentation, M was set to 200 –larger than the largest possible population size (i.e., the number of activity centres). The centroids of capture locations of individual animals caught were used as the starting values for activity centres, ensuring these occurred in suitable habitat as defined in the habitat mask. Three Markov Chain Monte Carlo (MCMC) chains with 60,000 iterations, a burn-in of 1,000, and a thinning rate of 10 were implemented. This combination of values ensured an adequate number of iterations to characterise the posterior distributions. Chain convergence was checked using the Gelman-Rubin statistic [44], R-hat, which compares between and within chain variation. R-hat values below 1.1 indicate convergence [45]. The approach of Royle et al. [46] was used for the model goodness-of-fit test, calculating three statistics, all using Freeman-Tukey discrepancies: individual animal by camera-station capture frequencies, aggregating the binary daily capture data by animals and camera-stations (FT1); individual animal capture frequencies, aggregating for each animal (FT2); and camera-station animal capture frequencies, aggregating for each camera-station (FT3).

Posterior distributions of adult male, female, overall population density and abundance, and male and female ranging parameters were generated from the model. Posterior locations of individual leopard activity centres were mapped, and animal density maps (individuals/km$^2$) for the two seasons were produced by modelling the movement of animals around activity centres.

## Site use

Leopard site use with respect to season, altitude and habitat was investigated using single-season, single-species occupancy modelling [47]. Detection histories, where '1' denoted a detection and '0' indicated a non-detection, were constructed using a five-day period as the sampling occasion. Bayesian occupancy analysis was performed in the program JAGS version 4.3.0 [48] accessed through the program R, version 3.6.0 [49], using the package RJAGS version 3–10 [50]. Three Markov chain Monte Carlo (MCMC) chains were run with 110,000 iterations, a burn-in of 10,000 and a thinning rate of 10. Chain convergence was checked with trace plots and the Gelman-Rubin statistic R-hat [44], which compares between and within chain variation. R-hat values below 1.1 indicate convergence [45]. Model fit was assessed using the Freeman-Tukey discrepancy measure [51]. Seasons were divided into a cool-wet winter and a hot-dry summer, altitude was calculated using a handheld GPS at each camera trap location, and habitat was classified into Fynbos and Karoo biomes [11].

## Livestock depredation

Reported livestock depredation data were obtained from CapeNature (the governmental organization responsible for biodiversity conservation in the Western Cape) for the period April 2009 to October 2018. This dataset covered the Cederberg region and surrounding districts, within 115 km from the study area, which is well within range of Cederberg leopard home ranges [11]. In R, we used the package 'lme4' to perform a generalized linear model with a Poisson distribution to test the effect of season on livestock depredation by leopard. Depredation events were grouped per month and multiple livestock killed in a single day (from the same area) were grouped into a single event. We set months nested within years as a random effect in the model due to the unequal number of repeated samples for each month.

## Results

### Sampling effort

Camera traps were active for 25,985 trap days (13,335 days in winter and 12,650 days in summer), with a total of 293,775 photographs obtained (excluding duplicate photographs from opposite cameras). Overall, 32 mammal species were photographed, four livestock species (goat, sheep, donkey, and cow), three domestic species (dog, cat, and horse), as well as various birds and reptiles (S2 Table). A total of 2,638 photographs were taken of leopards, which yielded 833 independent leopard captures (n = 435 in summer, n = 398 in winter). Leopards were photographed at 95% of camera trap stations (n = 69). We recorded an overall trap rate of 3.21 independent photographs per 100 days of sampling, and this was slightly higher in the summer (3.44) than in the winter (2.98).

In total, 63 different individual adult leopards were identified, consisting of 31 females, 26 males, and six of unknown sex. The accumulation curve for individual leopards in both summer and winter indicated that 80% of the individuals detected were counted within the first 100 days of trapping (Fig 3).

### Population size and density

The Bayesian model fitted well to the data for both seasons (FT1, FT2, FT3 P = 0.3–0.5), and R-hat values for all model parameters were below 1.1.

Leopard density was estimated as 1.53 (CV: 12.4%, 95% CI: 1.18–1.89) leopards/100 km$^2$ in winter and 1.62 (CV: 12.2%, 95% CI: 1.25–2.01) leopards/100 km$^2$ in summer (Table 1). Leopard density was more concentrated towards the central study area during summer, while in winter, leopard density was more spread-out across the study area with an increase in leopard density towards the periphery of the study area (Fig 4).

Leopard density was significantly higher for females than males, both in summer and winter (posterior probability = 1), with adult female to male ratio of 2.42:1 (95% CI: 1.34–3.55) for the summer period and 2.45:1 (95% CI: 1.53–3.47) for the winter period. The estimated mean leopard population size for the effectively sampled area based on the habitat mask was 113 (95% CI: 87–140) in summer and 107 (95% CI: 82–132) in winter.

### Site use

Leopard site use was high throughout the study area as leopards occurred at almost all camera trap sites. Site use was not affected by habitat type, Karoo: 0.9 (95% CI: 0.79–0.99) in summer and 0.9 (95% CI: 0.80–0.99) in winter, fynbos:0.84 (95% CI: 0.74–0.94) in summer and 0.88

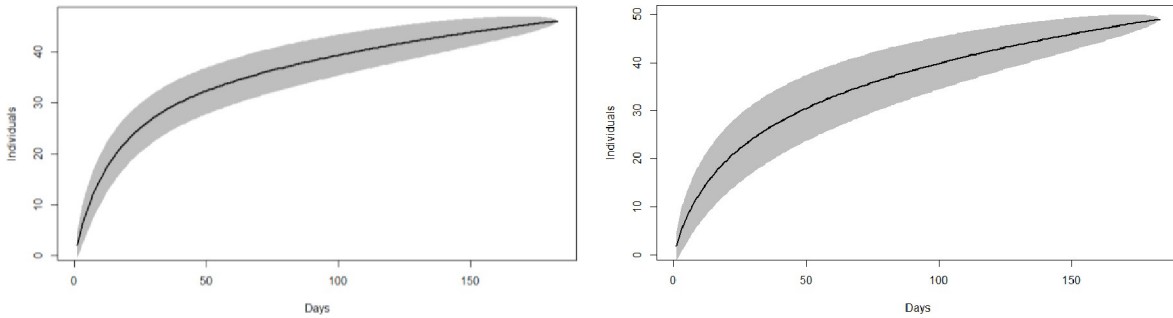

**Fig 3. Accumulation curve for individual leopards for summer (left) and winter (right).**

**Table 1. Estimates of the Bayesian spatial capture-recapture model parameters.** Sigma is the ranging scale parameter.

| Season | Density (95% CI) per 100 km² | Adult female adult density (95% CI) per 100 km² | Adult male density (95% CI) per 100 km² | Sigma (95% CI) (km) | Population size (95% CI) |
|---|---|---|---|---|---|
| Summer | 1.62 (1.25–2.01) | 1.14 (0.80–1.53) | 0.48 (0.34–0.63) | Female: 2.49 (2.21–2.78) | 113 (87–140) |
| | | | | Male: 4.92 (4.51–5.33) | |
| Winter | 1.53 (1.18–1.89) | 1.08 (0.75–1.43) | 0.45 (0.36–0.55) | Female: 3.11 (2.6–3.67) | 107 (82–132) |
| | | | | Male: 6.74 (6–7.5) | |

(95% CI: 0.80–0.96) in winter (S1 Fig). Leopards were also not influenced by altitude with site use remaining high (>0.8) along the altitudinal gradient during both seasons (S2 Fig).

## Ranging behaviour

Adult male movement parameter 'sigma' was significantly larger than the adult female sigma in both seasons (posterior probability = 1, Table 1). Male leopard sigma was also significantly larger in winter compared to the summer (posterior probability = 1). Activity centres of female leopards were mostly concentrated around the study core area in both seasons, while males displayed wider ranging behaviour with greater uncertainty in their activity centres (Figs 5 and 6). Based on the half-normal detection model [52], the estimated average home range radius was 12.05 km (95% CI: 11.05–13.06 km) for adult male leopards and 6.10 km (95% CI: 5.41–6.81 km) for adult female leopards in summer. In winter, average home range radius was 16.51 km (95% CI: 14.70–18.38 km) for males and 7.62 km (95% CI: 6.37–8.99 km) for females. Estimated home range size was 456 km² for males and 117 km² for females in summer and 856 km² for males and 182 km² for females in winter.

## Livestock depredation

A total of 222 livestock depredation events were recorded during the study period. Mean annual livestock depredation events was 22.2±10.4 [SD], and although events were lowest in

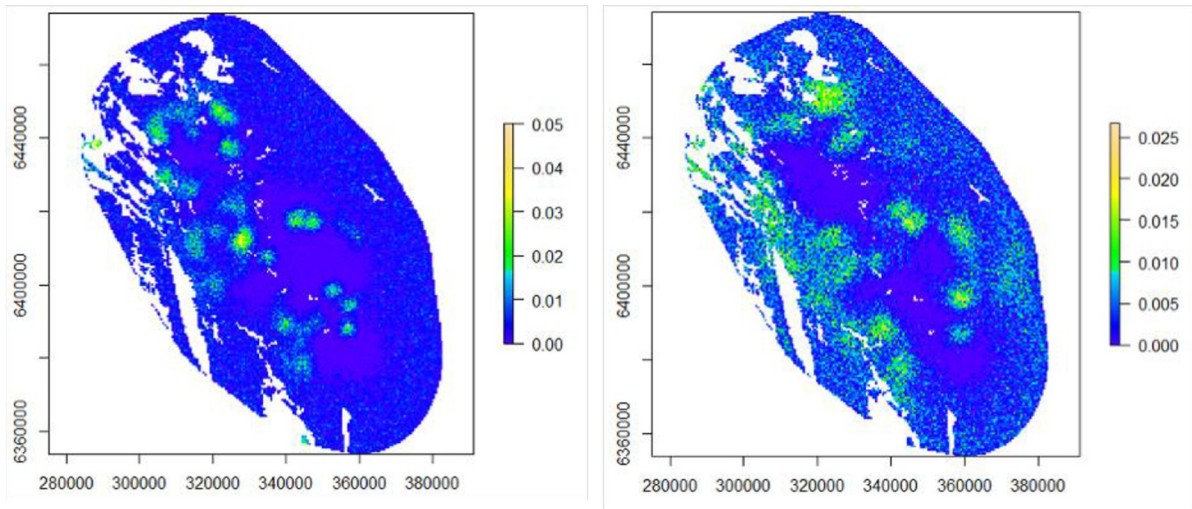

**Fig 4.** Expected density of individual leopards (individuals/km²) predicted from the Bayesian SCR model, for summer (left) and winter (right).

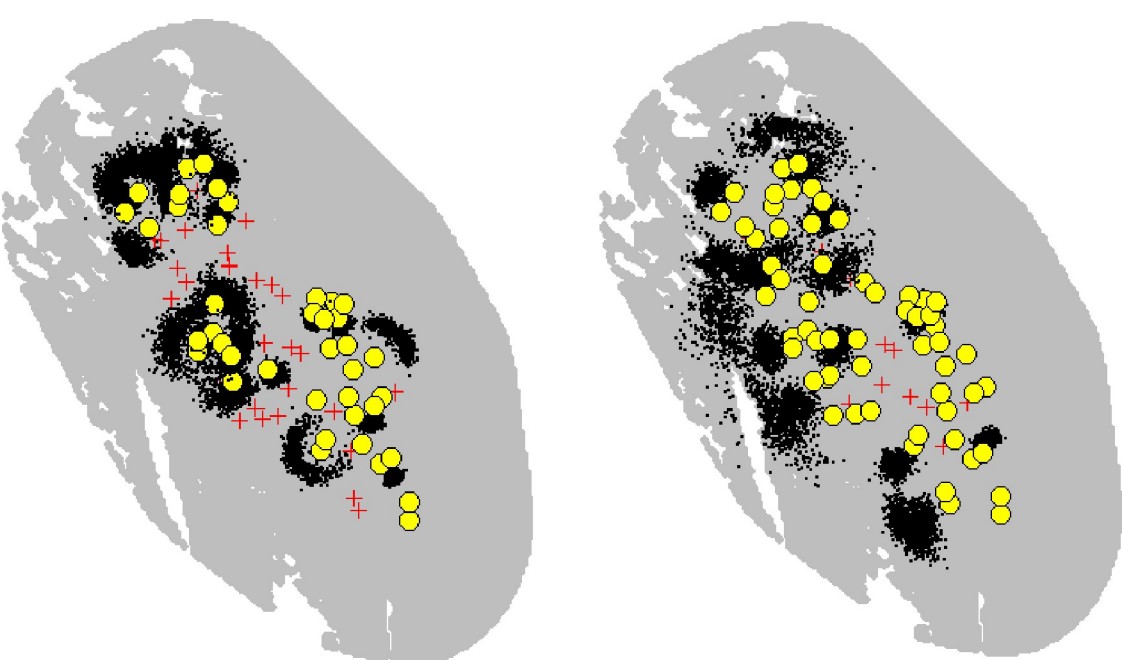

**Fig 5.** Activity centre posterior distributions (black dots); capture locations (yellow circles); and trap locations (red crosses) for all recorded adult females (left) and adult males (right), in the summer.

2009 (n = 3) and highest in 2018 (n = 42), exploratory analysis did not indicate an increasing or decreasing trend in livestock depredation over time. The number of reported livestock depredation events by leopards differed significantly between seasons (Z = 3.3, p < 0.01).

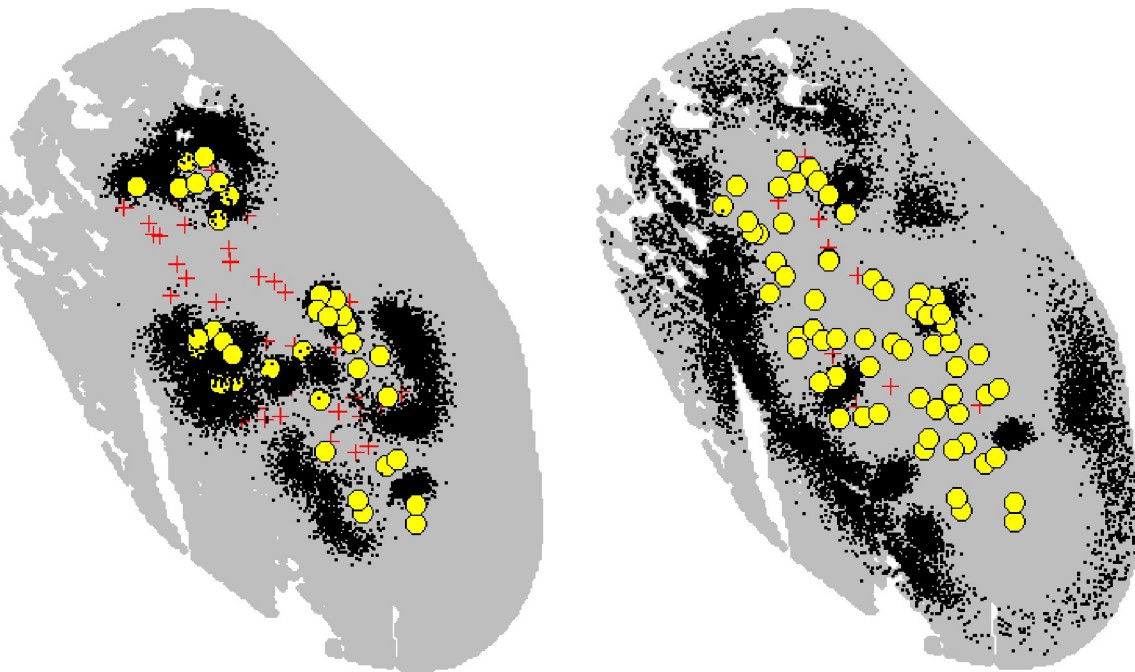

**Fig 6.** Activity centre posterior distributions (black dots); capture locations (yellow circles); and trap locations (red crosses) for all recorded adult females (left) and adult males (right), in the winter.

Mean monthly depredation events reported during summer were 1.4 (max = 4) and 2.4 (max = 10) during winter, while mean number of livestock killed seasonally was 7.7 (range = 2–12) during summer and 14.9 (range = 1–38) during winter.

## Discussion

Robust population estimates are key requirements for the effective conservation of threatened species [18]. Previous attempts to estimate leopard densities in the Cederberg were based on a variety of methods, which were both limited and yielded wide-ranging estimates [11, 27, 28]. Our study provides the first methodologically and statistically rigorous survey of leopards in the Cederberg by using a Bayesian SCR method on an extensive camera trapping survey. Given the mounting threats facing leopards throughout their range, our study has provided valuable insights into the status and spatial distribution of this elusive species across such a largely human-dominated landscape.

Our study revealed a relatively low mean leopard density of 1.53–1.62 leopards/100 km$^2$, which corresponds with density estimates from other leopard populations in the Western and Eastern Cape Provinces of South Africa [11, 14, 16, 53]. Therefore, along with these studies, our data suggest that leopards occur at a low density (<2 leopards/100 km$^2$) throughout the region. Although the methodologies used to calculate leopard densities between this study and Martins' study [11] differed (SCR vs. telemetry data), the similar densities suggest that the Cederberg leopard population has remained relatively stable for at least the past decade. Comparing densities using two different methods is generally not advised, however Devens et al. [16] found that little variation exists between density estimates obtained from SECR and GPS/telemetry methods. Our estimates were also comparable to leopard densities from other semi-arid environments in southern Africa [54, 55].

Leopards in the Cape occur in a highly modified and fragmented landscape and their low densities are likely a consequence of several bottom-up and top-down processes. Firstly, prey availability is typically an important determinant of leopard density, as lower prey densities correlate with lower leopard densities [54, 56, 57]. In the Cederberg, leopards are exposed to a limited and small-bodied prey base [31], which is a consequence of low productivity in the area, and this could explain the low leopard densities found here and throughout the Cape region. Secondly, human-related activities may result in reduced leopard densities and recent studies suggest that anthropogenic disturbance may exert even stronger pressures on leopard space use compared to prey availability [58, 59]. Leopards in the Cape Provinces mainly occur in fragmented protected areas which are surrounded by landscapes that are largely unconducive to their survival. Moreover, these leopard populations have experienced long-term persecution (both legal and illegal) with records of hundreds of leopards being killed within relatively short periods [60]. Within a two-year period (1988–1990), 21 leopards were legally killed as 'problem animals' in the Cederberg area through permits issued by Cape Nature (excluding illegal killing), and such off-take may well disrupt leopard population dynamics within the system [11]. Persecution of leopards still occurs in and around the study area, but at a comparatively smaller scale e.g. only two leopards were killed in six-year period (2004–2010). This was attributed to improved awareness and conservation efforts in the area [11]. Thirdly, leopard spatial distribution may be influenced by interspecific competition with dominant carnivores [57]. However, as the only remaining large carnivore in the region, these leopards have no direct competitors and are thus not spatially restricted by heterospecifics [61].

To estimate ranging behaviour, leopards are typically fitted with satellite and/or VHF collars. However, in this study we demonstrate how individual leopard activity centres were constructed using camera trapping and SECR framework. The ranging scale parameter was larger

for both sexes in the winter, which coincides with the larger home ranges recorded in winter by Martins [11]. This technique can be applied to other carnivore species that are individually recognisable and may be useful to determine ranging behaviour without the need for invasive and costly collaring.

In areas where leopards occupy large home ranges and occur at low densities, effectively conserving viable populations within the limits of protected areas is a major challenge [56]. Leopards moving beyond the 'safe zone' of protected areas become vulnerable to edge effects which can have significant impacts on population densities [17]. For example, the removal of resident leopards from non-protected areas can generate vacant gaps in the landscape, luring leopards from protected areas and leading to a 'vacuum effect' [17]. This may even affect carnivores at the very core of large protected areas [59, 62]. Our study revealed seasonal shifts in male leopard activity centres away from the protected area and into the surrounding farmland matrix during winter. These shifts could be linked to seasonal resource availability [63]. Livestock birthing peaks during the winter months (April–June), may present an attractive and easy alternative food source for leopards in farmlands. However, other factors such as leopard breeding season may also influence seasonal leopard movement within a landscape [64]. Interestingly, reported livestock depredations were significantly higher in winter compared to summer, which aligns with leopard movement into the surrounding farmland matrix, and also corresponds with the findings of Stuart [65]. Despite this, it is likely that the non-protected (privately-owned) land may play a critical role in supporting leopard home ranges [14] and their energetic requirements by providing alternative food sources during winter. Interestingly, dietary studies have revealed that livestock contributes only a fraction towards overall leopard diet (e.g. [31]), although these studies were largely conducted within the protected area.

We found that reported livestock depredation events did not show a particular trend over time. This could be due to inconsistent reporting as data collected is reliant upon landowners reporting their losses to the relevant authorities and if landowners are not incentivised, reporting might not be their priority. This observed variation in depredation events over the years could also be influenced by leopard occupancy across the landscape over time [66]. Nevertheless, fewer leopards are killed annually compared to before conservation efforts commenced in 2008 [11] and so engaging with landowners to improve tolerance towards leopards and reducing livestock depredations through mitigation measures may be key in promoting human-leopard coexistence in the region.

Inadequate livestock guarding practices have regularly been cited as a significant contributor to livestock depredation events (e.g. [67]). Good husbandry practices are particularly important in winter when livestock birthing peaks, livestock attacks are highest, and leopards venture into human-dominated landscapes. Therefore, based on our findings, we identify winter as a critical period in the Cederberg for both the farmer and the leopard. We encourage researchers and protected area managers to partner with livestock producers in the surrounding farmlands to quantify the effect of various conflict mitigation strategies and to generate evidence-based approaches that will ultimately improve our capacity to reduce livestock depredation events and prevent retaliations against leopards [68].

Occupancy results indicate that leopards are homogenously distributed across the study area, being detected at 95% of camera trapping sites, which essentially covered the entire Cederberg protected area landscape including the Fynbos and Karoo biomes. Due to their broad distribution, leopard occupancy was largely unaffected by habitat, season, or altitude, as found by Martins & Harris [29], highlighting their adaptability and ecological plasticity. Studies show that leopard spatial distribution is negatively correlated with human encroachment around protected areas [17, 55, 58, 69]. This vulnerability is relevant to leopard conservation

as they typically display higher resilience to human-related activities compared to other large carnivores [59]. A previous study from the Cederberg found that leopard space use was not influenced by human settlements [29], although this study was also focused within the Cederberg protected area where human activity is relatively low and persecution is negligible.

An emerging threat to leopards in the Western Cape is illegal poaching of bushmeat [70]. Wire snaring is a highly effective method for trapping small to medium-sized mammals (the intended targets); however, snares are indiscriminate and leopard mortalities due to snares are known to occur. In 2019, the Cape Leopard Trust (an NGO focused on leopard research and conservation) initiated anti-snare patrols in the Boland region of the Western Cape in an attempt to map, quantify and control this threat. At present, snaring does not seem to be prominent in the Cederberg, although we do not disregard poaching as a potential threat. Poaching can result in the depletion of both predator and prey populations [58], so it is critical to monitor and control this threat. Recent studies show that drastic short-term declines in leopard populations may occur due to anthropogenic influences (e.g. [33]). It is therefore recommended that repeated studies are conducted every few years (2–3 years) using standardised SECR methods to monitor population trends and identify these changes in the population timeously and address the issues causing these declines using effective conservation intervention measures to reverse these declines [17, 71].

Implementation of conservation measures targeted at reducing edge effects and promoting conservation of leopards are required to ensure the long-term persistence of leopards in a human-dominated landscape. For example, through the focused research between 2004 and 2010, significant steps have already been taken to improve conservation efforts of leopards in the Cape, including: (1) abandoning the relocation of 'problem animals', as this practice may result in more harm than good, (2) reducing the duration of permits issued by Cape Nature to destroy 'problem leopards' from one month to seven days, and (3) stopping the removal of dominant leopards, as the vacancy created may soon be occupied by a new, immigrant leopard (see Martins [11]). Furthermore, increasing protected area size is an effective way of reducing edge effects and aiding wildlife conservation [17, 59, 72]. The Western Cape is largely modified and the remaining suitable leopard habitat (49,850 km$^2$ or 38% of the province) is highly fragmented with current PAs comprising only one third (15,010 km$^2$) of this suitable habitat [8]. The largest proportion of suitable leopard habitat remains outside of protected areas and thus leopard conservation efforts should focus on these private, formally non-protected lands [8]. This could be achieved through stewardship sites (i.e. individual landowners committing to protect and manage their properties or parts thereof according to sound conservation management principles under the guidance and support of CapeNature) or the formation and expansion of conservancies (i.e., collaborative management of privately-owned farms to conserve biodiversity and remove lethal predator control as a management tool). Well managed conservancies may result in increased wildlife populations as well as greater species resilience to environmental stochasticity [73], and conservancy members may have higher tolerance to large carnivores compared to non-conservancy members [74]. The Cederberg Conservancy was established in partnership with CapeNature in 1997 as a voluntary agreement between landowners. It currently consists of 22 privately-owned properties and covers 755 km$^2$, resulting in more viable habitat for leopards and their prey. The size of this conservancy could certainly be increased to the benefit of the leopard population.

Landscape connectivity between the various suitable-leopard habitats is another key aspect which will certainly impact the future of leopard conservation in the Cape. Anecdotal evidence suggests that at least some of the leopard 'subpopulations' within the Western Cape may be interconnected. This poses several important considerations to maintaining a genetically viable leopard population in the long term. Ultimately, maintaining and potentially expanding

buffer zones around protected areas and dispersal corridors among suitable leopard areas is key to the long-term sustainability of South Africa's leopard populations [8].

## Conclusion

Effectively conserving leopards in a human-dominated landscape requires a good understanding of population estimates. The robust statistical approaches used in our study are important as they provide accurate population estimates that can be used to make informed management and conservation decisions at both local and regional scales. Low leopard density and large ranging behaviour indicate that large tracts of protected land and connectivity of the landscape, as well as increased conflict mitigation efforts, especially during winter months when male leopard activity centres shift towards surrounding farmland and livestock depredation events increase, are essential requirements for the survival of this leopard population [55]. Our study found that leopards were homogenously distributed across the Cederberg study site and were unaffected by altitude, habitat or season. However, we recommend future studies to quantify the density gradient of leopards between protected and non-protected areas to evaluate the anthropogenic impact on leopard populations. Protected areas play an important role in the conservation of leopard populations, although the majority of the remaining suitable leopard habitat in South Africa falls within non-protected areas [8]. Therefore, it is crucial that buffer zones around protected areas and dispersal corridors among areas with suitable habitat are maintained, which will also require human-wildlife conflict mitigation, to ensure the long-term conservation of the South African leopard population [8]. We encourage future research to investigate the potential interconnectivity between the South African leopard populations and whether the corridor networks require protection and management.

## Supporting information

**S1 Table. Details of camera trap locations in the Cederberg study area.**
(XLSX)

**S2 Table. Independent photos for each species captured during the camera trapping survey.** All animals were identified to species level, while except for birds and reptiles which were grouped.
(XLSX)

**S1 Fig. Leopard occupancy in Fynbos and Karoo habitat during summer and winter.**
(TIF)

**S2 Fig. Occurrence of leopards at difference altitude ranges during summer and winter.**
(TIF)

**S3 Fig.**
(JPG)

## Acknowledgments

We are very grateful to Abax Foundation, Bushmans Kloof Wilderness Reserve, Ford Wildlife Foundation, Lomas Wildlife Protection Trust, and the Rolf Stephan Nussbaum Foundation for their support. A special thanks also to Rika du Plessis from Cape Nature and all Cederberg landowners for their cooperation and providing us access to their properties. We acknowledge the Cape Leopard Trust for facilitating this research and the Scientific Advisory Board for their input during the design phase of this study. We thank Gareth Mann for his guidance on

selecting good camera trap sites for leopards and his assistance with the Camerabase software. We thank Ewan Brennan, Hannes de Kok, Ismail Wambi, Lizette Burger, James Wells, and Grant Baker for assisting with the camera trap survey and MJ Grobler, Louw Redelinghuys, and Ockie Müller for assisting with the data processing. Finally, we want to thank Professor Dan Parker and Dr Alison Leslie for reviewing this manuscript.

## Author Contributions

**Conceptualization:** Lana Müller, Christiaan Stefanus Lochner, Rajan Amin.

**Data curation:** Lana Müller, Barbara Catharine Seele, Christiaan Stefanus Lochner.

**Formal analysis:** Christiaan Stefanus Lochner, Rajan Amin.

**Funding acquisition:** Lana Müller.

**Investigation:** Lana Müller, Christiaan Stefanus Lochner.

**Methodology:** Lana Müller, Rajan Amin.

**Project administration:** Lana Müller.

**Supervision:** Lana Müller.

**Validation:** Rajan Amin.

**Visualization:** Christiaan Stefanus Lochner, Rajan Amin.

**Writing – original draft:** Lana Müller, Willem Daniel Briers-Louw, Rajan Amin.

**Writing – review & editing:** Lana Müller, Willem Daniel Briers-Louw, Barbara Catharine Seele, Christiaan Stefanus Lochner, Rajan Amin.

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
