## [Decision Letter · Decision Letter 0]

1 Oct 2021

PONE-D-21-20635Population size, density, and ranging behaviour in a key leopard population in the Western Cape, South AfricaPLOS ONE

Dear Dr. Müller,

Thank you for submitting your manuscript to PLOS ONE. After careful consideration, we feel that it has merit but does not fully meet PLOS ONE’s publication criteria as it currently stands. Therefore, we invite you to submit a revised version of the manuscript that addresses the points raised during the review process.

We look forward to receiving your revised manuscript.

Kind regards,

Bi-Song Yue, Ph.D

Academic Editor

PLOS ONE

3. In your Methods section, please provide additional information regarding the field permits you obtained for the work. Please ensure you have included the full name of the authority that approved the field site access and, if no permits were required, a brief statement explaining why.

“We are very grateful to our donors, Abax Foundation, Bushmans Kloof Wilderness Reserve, Lomas Wildlife Protection Trust, and the Rolf Stephan Nussbaum Foundation. Without their support this work would not have been possible. We thank Ford Wildlife Foundation for providing a vehicle for the research team. A special thanks also to Rika du Plessis from Cape Nature and all Cederberg landowners for their support and providing us access to their properties. We acknowledge the Cape Leopard Trust for providing the resources to enable this research and the Scientific Advisory Board for their input during the design phase of this study. We thank Gareth Mann for his guidance on selecting good camera trap sites for leopards and his assistance with the Camerabase software. We thank Ewan Brennan, Hannes de Kok, Ismail Wambi, Lizette Burger, James Wells, and Grant Baker for assisting with the camera trap survey and MJ Grobler, Louw Redelinghuys, and Ockie Müller for assisting with the data processing. Finally, we want to thank Professor Dan Parker and Dr Alison Leslie for reviewing this manuscript.We note that you have provided funding information that is not currently declared in your Funding Statement. However, funding information should not appear in the Acknowledgments section or other areas of your manuscript. We will only publish funding information present in the Funding Statement section of the online submission form. “

“The work is supported by the Cape Leopard Trust whom received funding from Abax Foundation, Bushmans Kloof Wilderness Reserve, Lomas Wildlife Protection Trust, and the Rolf Stephan Nussbaum Foundation. Ford Wildlife Foundation has sponsored a vehicle for the research team.”

6. We note that Figure 1 in your submission contain [map/satellite] images which may be copyrighted. All PLOS content is published under the Creative Commons Attribution License (CC BY 4.0), which means that the manuscript, images, and Supporting Information files will be freely available online, and any third party is permitted to access, download, copy, distribute, and use these materials in any way, even commercially, with proper attribution. For these reasons, we cannot publish previously copyrighted maps or satellite images created using proprietary data, such as Google software (Google Maps, Street View, and Earth). For more information, see our copyright guidelines: http://journals.plos.org/plosone/s/licenses-and-copyright.

In the figure caption of the copyrighted figure, please include the following text: “Reprinted from [ref] under a CC BY license, with permission from [name of publisher], original copyright [original copyright year].

Reviewers' comments:

Reviewer's Responses to Questions

**Comments to the Author**

1. Is the manuscript technically sound, and do the data support the conclusions?

Reviewer #1: Yes

Reviewer #2: Yes

2. Has the statistical analysis been performed appropriately and rigorously? 

Reviewer #1: Yes

Reviewer #2: Yes

3. Have the authors made all data underlying the findings in their manuscript fully available?

Reviewer #1: No

Reviewer #2: Yes

4. Is the manuscript presented in an intelligible fashion and written in standard English?

Reviewer #1: Yes

Reviewer #2: Yes

5. Review Comments to the Author

Reviewer #1: I think that it is an interesting contribution that adds to the knowledge of leopard’ populations at low densities. The results obtained with telemetry are similar, so the information obtained with remote photography can be regarded as plausible. The method used is less laborious than the traditional telemetry. Authors used the most innovative techniques for data analyses (Bayesian statistics), combining SCR methods for calculating population and species density, and occupancy models to investigate the effect of some predictors on the species distribution. I have not very much to say… Please, find some minor comments attached.

Fig.1, Please, include a reference map of Africa continent

Give the average range size for males and females.

If you divided the area in 50 km2 grids, it seems that every cell will have (if square) a 7km side; I do not understand an interdistance of 2 km between cameras? It is non-sense to put more cameras in some cells, which will increase detectability in cells with more cameras. Please, justify. Include the grid in fig.1. Also provide a mean interdistance and a range between cameras, not only the minimum distance.

Cameras were operative 24h, but how many days were in the field? The 73 cameras were simultaneously placed from October 2017 to September 2018 (11 months)? Please, clarify.

SCR? Needs to be described at first citation.

Line 170: Why these periods were selected, justify or add some reference. Where the curves of Fig.2 asymptotic? Can you calculate the actual proportion of the population sampled if the curves reached an asymptote?

Line 206: Why a 5 days-period? A weekly interval would have more sense, justify.

I guess that “depredation” is not a correct term applied to the ecology of organisms, maybe better using “predation”

Line 217: I think that a more interesting question (rather than seasonal predation) is whether predation events increased/decreased along the study period (2009-2018). This was not analysed, and it should be. Maybe if decreasing, this could be a reason to lower persecution and should be discussed.

Line 221: Despite the use of Poisson distribution can be right, maybe the low number of predation events (a lot of zeroes) can produce underdispersion issues. The use of the negative binomial sometimes is a better solution, but this might be checked.

Line 229: I think that is necessary to have all the records obtained available to the readers (photos/contacts per species). Please, include it in supplementary material.

Line 231: Numbers < 10 in words.

Line 235: mean trap rates need some dispersal statistics

Reviewer #2: I reviewed the manuscript “POPULATION SIZE, DENSITY AND RANGING BEHAVIOR IN A KEY LEOPARD POPULATION IN THE WESTERN CAPE, SOUTH AFRICA”. The manuscript certainly has merit for publication. The manuscript certainly has merit for publication with minor improvement on the results by including camera trap locations(GPS coordinates) and conclusion by including part of missing findings.

6. PLOS authors have the option to publish the peer review history of their article (what does this mean?). If published, this will include your full peer review and any attached files.

Reviewer #1: No

Reviewer #2: No

---

## [Author Response · Author response to Decision Letter 0]

12 Jan 2022

Please see the attached Response to Reviewers Letter

---

## [Editor Report · Decision Letter 1]

21 Apr 2022

Population size, density, and ranging behaviour in a key leopard population in the Western Cape, South Africa

PONE-D-21-20635R1

Dear Dr. Müller,

We’re pleased to inform you that your manuscript has been judged scientifically suitable for publication and will be formally accepted for publication once it meets all outstanding technical requirements.

Kind regards,

Stephanie S. Romanach, Ph.D.

Academic Editor

PLOS ONE
---

## [Editor Report · Acceptance letter]

19 May 2022

PONE-D-21-20635R1 

Population size, density, and ranging behaviour in a key leopard population in the Western Cape, South Africa 

Dear Dr. Müller:

I'm pleased to inform you that your manuscript has been deemed suitable for publication in PLOS ONE. Congratulations! Your manuscript is now with our production department. 

Kind regards, 

on behalf of

Dr. Stephanie S. Romanach 

Academic Editor

PLOS ONE